# The Influence of Ultra-Low Tidal Volume Ventilation during Cardiopulmonary Resuscitation on Renal and Hepatic End-Organ Damage in a Porcine Model

**DOI:** 10.3390/biomedicines11030899

**Published:** 2023-03-14

**Authors:** Katja Mohnke, Victoria Buschmann, Thomas Baller, Julian Riedel, Miriam Renz, René Rissel, Alexander Ziebart, Erik K. Hartmann, Robert Ruemmler

**Affiliations:** Medical Center, Department of Anaesthesiology, Johannes Gutenberg University, Langenbeckstr. 1, 55131 Mainz, Germany

**Keywords:** CPR, ventilation, circulation, renal damage, liver damage

## Abstract

The optimal ventilation strategy during cardiopulmonary resuscitation (CPR) has eluded scientists for years. This porcine study aims to validate the hypothesis that ultra-low tidal volume ventilation (tidal volume 2–3 mL kg^−1^; ULTVV) minimizes renal and hepatic end-organ damage when compared to standard intermittent positive pressure ventilation (tidal volume 8–10 mL kg^−1^; IPPV) during CPR. After induced ventricular fibrillation, the animals were ventilated using an established CPR protocol. Upon return of spontaneous circulation (ROSC), the follow-up was 20 h. After sacrifice, kidney and liver samples were harvested and analyzed histopathologically using an Endothelial, Glomerular, Tubular, and Interstitial (EGTI) scoring system for the kidney and a newly developed scoring system for the liver. Of 69 animals, 5 in the IPPV group and 6 in the ULTVV group achieved sustained ROSC and were enlisted, while 4 served as the sham group. Creatinine clearance was significantly lower in the IPPV-group than in the sham group (*p* < 0.001). The total EGTI score was significantly higher for ULTVV than for the sham group (*p* = 0.038). Aminotransferase levels and liver score showed no significant difference between the intervention groups. ULTVV may be advantageous when compared to standard ventilation during CPR in the short-term ROSC follow-up period.

## 1. Introduction

There are no specific recommendations for optimal ventilation during cardiopulmonary resuscitation (CPR). Neither the European Resuscitation Council (ERC) nor the American Heart Association (AHA) provide detailed recommendations for ventilation during CPR. Provided that trained personnel are present, these associations recommend that the airway is secured by endotracheal intubation, allowing for continuous chest compressions. Meanwhile, ventilation should be performed at a rate of 10 ventilation strokes per minute [1,2]. Such imprecise specifications, however, lead to insufficient guideline adherence and varying ventilation strategies during CPR [3]. 

Even so, endotracheal intubation could facilitate differentiated ventilation strategies to benefit critically ill patients suffering from acute respiratory distress syndrome (ARDS) [4,5]. In a previous study, Ruemmler et al. [6] showed that ultra-low tidal volume ventilation (ULTVV) with tidal volumes of 2–3 mL kg^−1^ had no adverse effects on oxygenation and ventilation and showed a lower cerebral cytokine expression [6], indicating reduced end-organ damage.

Nevertheless, it has not yet been clarified how ULTVV specifically affects renal and hepatic function and organ integrity during resuscitation in the short-term after return of spontaneous circulation (ROSC). Since acute kidney injury and hepatic hypoxia following CPR play a relevant role in morbidity and mortality in post-resuscitation treatment [7,8], it is crucial to elucidate the effects of ULTVV. Therefore, this study aims to validate the hypothesis that ULTVV during cardiopulmonary resuscitation mitigates the renal and hepatic end-organ damage in the short-term follow-up period after ROSC when compared to standard ventilation (Intermittent Positive Pressure Ventilation (IPPV) with tidal volumes of 8–10 mL kg^−1^. Hence, early effects of ULTVV and IPPV on hemodynamic, functional, and histopathological parameters of the kidney and the liver in a porcine CPR model were compared using adapted tissue damage scoring systems for both organs.

## 2. Materials and Methods

### 2.1. Anesthesia and Instrumentation

The study was conducted after approval of the State and Institutional Animal Care Committee (Landesuntersuchungsamt Rheinland-Pfalz, Koblenz, Germany; G 16-1-042) from 28 May 2019 to 22 April 2020 and in accordance with the ARRIVE guidelines. Here, 69 juvenile, predominantly male, pigs (*Sus scrofa domestica*; mean weight 30 ± 3 kg; age: 12 weeks) were acquired from a local breeder. After intramuscular injection of ketamine (Hameln Pharmaceuticals GmbH, Hameln, Germany; 1.5 mg kg^−1^), azaperone (Lilly Deutschland GmbH, Bad Homburg, Germany; 2.5 mg kg^−1^), and midazolam (Hameln Pharmaceuticals GmbH, Hameln, Germany; 0.3 mg kg^−1^), the sedated animals were delivered to the laboratory. 

After placement of an IV access to the ear, anesthesia was induced with a bolus injection of fentanyl (Janssen-Cilag, Neuss, Germany; 4 µg kg^−1^) and propofol (Fresenius Kabi, Bad Homburg, Germany; 2 mg kg^−1^). A single dose of atracurium (HEXAL AG, Holzkirchen, Germany; 0.5 mg kg^−1^) was administered to facilitate endotracheal intubation. After endotracheal intubation, the animals were ventilated using an intensive care respirator (Engstroem care station, GE healthcare, Munich, Germany) in volume-controlled mode (tidal volume 6 mL kg^−1^, positive end-expiratory pressure of 5 mbar, FiO_2_ of 0.4, inspiration to expiration ratio 1:2) and variable respiration rate to achieve an end-tidal CO_2_ < 6 kPa. 

Throughout the entire experiment, anesthesia was maintained using a continuous infusion of fentanyl (0.1–0.2 mg h^−1^) and propofol (8–12 mg kg^−1^ h^−1^).

With ultrasound guidance, arterial and central venous access was established in the inguinal region as described in previous studies by our group [9]. A pulse contour cardiac output system (PiCCO, Pulsion Medical Systems, Munich, Germany), central venous catheter, and a fibrillation catheter (VascoStim B 2/5F, Vascomed GmbH, Binzen, Germany) were established. 

The animals initially received an intravenous infusion of an isotonic electrolyte solution of 30 mL kg^−1^ (Sterofundin^®^, B. Braun Melsungen AG, Melsungen, Germany), followed by a continuous infusion of 5 mL kg^−1^ h^−1^. In the case of ROSC, an intravenous infusion of 30 mL kg^−1^ of this electrolyte solution was administered over a 2 h period; subsequently, the hourly running rate was halved to 2.5 mL kg^−1^.

### 2.2. Intervention

After instrumentation and 30 min of consolidation, baseline measurements were documented. Five animals were monitored exclusively (sham group). 

Ventricular fibrillation was induced via the fibrillation catheter as previously described [9] in the remaining animals, whereby ventilation was disconnected during monitor-confirmed cardiac arrest. After eight minutes without any treatment, the animals were randomized by pulling sealed envelopes as described in Table 1. 

During the intervention, both groups received mechanical chest compressions for eight minutes using a LUCAS-2^®^ device (PhysioControl^®^, Lund, Sweden) with a fixed compression rate of 100 min^−1^, accompanied by the allocated ventilation intervention. After eight minutes of basic life support, the animals received adapted advanced life support for another ten minutes: two minute compression cycles, rhythm analysis and defibrillation if indicated (200 J, bi-phasic, electrodes in anterior-lateral position), including the administration of epinephrine (1 mg) and vasopressine (0.1 U kg^−1^) after the 1st, 3rd, 6th, and 9th unsuccessful defibrillation attempts and amiodarone (150 mg) after the 3rd and 6th unsuccessful defibrillation attempts. If ROSC was not achieved after ten defibrillations, the experiment was terminated and those animals were excluded from further data analysis. Animals with ROSC were returned to standard ventilation as described at baseline, followed by a 20 h monitoring period. The goal was to achieve a peripheral oxygen saturation greater than 93%. If necessary, the ventilation invasiveness was adjusted according to the specifications of the Acute Respiratory Distress Syndrome (ARDS)-network [10]. Mean arterial blood pressure was kept above 60 mmHg using norepinephrine administration and volume boluses. 

The experiment was terminated by injecting 40 mmol potassium chloride via the central venous catheter after deepening general anesthesia with 200 mg propofol.

### 2.3. Measurements/Sample Collection 

Hemodynamic and respiratory parameters were continuously measured and recorded throughout the experiment using the Datex Ohmeda S5 monitor (GE Healthcare, Munich, Germany) and the Engström Carestation, respectively. Arterial and central venous blood gas analyses were performed at baseline, at ROSC, and hourly after ROSC using a blood gas analyzer (Radiometer: BGA ABL 90 Flex, Radiometer, Krefeld, Germany). 

Further laboratory data, which included creatinine, alanine aminotransferase levels (ALT), and aspartate aminotransferase levels (AST), were taken at baseline, 6, and 20 hours after ROSC. Creatinine clearance (by means of urine collected at the end of the observation period) was calculated accordingly.

After ROSC, a suprapubic bladder catheter was placed under sonographic guidance, through which urine was drained and collected until the end of the experiment. These samples were examined by the central laboratory of the University Medical Center Mainz. After termination, tissue samples of the left kidney and of the left lobe of the liver were harvested and fixed with formalin 4%. The area between the central renal pelvis and the upper renal pole was excised from the kidney. These tissue samples were then embedded in paraffin, cut into 2 µm thick slices, and stained with hematoxylin eosin (HE) stain by the Department for General Pathology of the University Medical Center Mainz. Tissue samples were provided by the tissue bank of the University Medical Center Mainz in accordance with the regulations of the tissue bank. The histological samples were examined microscopically in an investigator-blinded manner using an Olympus microscope (CX43RF, Olympus Corporation, Tokyo, Japan) with Olympus cellSens Entry Software (Olympus cellSens Entry, Version 2.1, Olympus Corporation, Tokyo, Japan).

### 2.4. Tissue Damage Scoring Systems 

For evaluating kidney damage, the established EGTI-scoring system was adapted and applied. This system comprised the following items: endothelial, glomerular, and tubulointerstitial tissue damage with scores ranging from 0 = no damage to 5 = maximum tissue damage per subcategory [11].

Moreover, liver damage was determined using a newly developed pathology scoring system consisting of centrilobular necrosis, hydropic cell swelling, inflammation, venous congestion, and sinusoidal congestion with scores ranging from 0 = no damage to 5 = maximum tissue damage per subcategory (Appendix A). Since there were neither models for the assessment of porcine livers in general nor for histopathological changes in the liver after CPR, the scoring system was based on established liver scoring systems that focus on liver fibrosis [12,13,14]. These are, among other things, based on the metabolic zones of the liver acini [15]. The existing scoring systems were adapted to the expected histopathological damage after CPR, specifically the consequences of the venous congestion and the hypoxemic undersupply of the tissue.

### 2.5. Statistics 

Categorical variables were tested with Pearson’s Chi-square test [16]. Continuous variables were evaluated for normality using the Shapiro–Wilks test [17] and homogeneity for variances using Levene’s test [18]. For single measurements, if normal distribution was not given, the Mann–Whitney–U test [19] or Kruskal–Wallis test [20] was used as appropriate. Statistical analyzes were performed using 2-way ANOVA inter-group tests with post-hoc Bonferroni correction for repeated measures [21], if normal distribution was given. Otherwise, the Mann–Whitney–U test was used. Statistical evaluation was performed using IBM SPSS Statistics (IBM SPSS Statistics for Windows, Version 20. IBM Corporation, Armonk, NY, USA). The data in the text are presented as mean (standard deviation). Bar plots are presented as mean with the standard error of the mean. A significance level of 0.05 was set.

## 3. Results

69 experiments were performed. A total of 54 animals were excluded. Most (*n* = 43) were excluded per protocol after no ROSC was achieved after the 10th defibrillation attempt. 11 animals were excluded because of relevant pre-existing medical conditions, equipment failure, or fatal injury from the resuscitation effort. 

ROSC was achieved in five of the IPPV animals and six of the ULTVV animals (*p* = 0.402). Four animals could be included as the sham comparison group. Table 2 summarizes the hemodynamic data and shows no differences between the groups at baseline.

### 3.1. Hemodynamic Stability

There was no difference between the intervention groups in terms of post-resuscitation norepinephrine requirements. 

Lactate levels showed significant differences at the time points T0 (ROSC) to T5 between the Sham and ULTVV (*p* < 0.029) and between the Sham and IPPV (*p* < 0.026), but no differences between the intervention groups (Figure 1a). T5 denotes five hours after ROSC, T20 denotes twenty hours after ROSC, etc.

The central venous pressure showed no significant group differences over the course of the experiment (*p* = 0.517) (Figure 1b).

### 3.2. Function and Histopathological Damage of the Kidney

Serum creatinine levels increased in all animals over the course of the experiment (*p* < 0.001), with no group differences discerned (*p* = 0.478) (Figure 2a). Creatinine clearance of the IPPV group was significantly lower than that of the sham group (*p* < 0.001); this was not the case for the ULTVV group vs. Sham (*p* = 0.077). There was no significant difference between the intervention groups (*p* = 0.491) (Figure 2b). Urea levels increased over the course of the experiment (*p* = 0.001), with no significant group difference (*p* = 0.378). The amount of urine excreted over the entire follow-up period showed no significant group difference (*p* = 0.447).

Postmortem analysis of kidney tissue shows a significantly higher EGTI total score for ULTVV vs. Sham (*p* = 0.038); for IPPV vs. Sham there was no significant difference (*p* = 0.063). There was no significant difference between the intervention groups (*p* = 0.931) (Figure 3a). For the individual items of the EGTI score, there were no significant differences between the intervention groups (Figure 3b). 

### 3.3. Function and Histopathological Damage of the Liver

Alanine Aminotransferase levels (ALT) as a function of time were elevated in all groups (*p* = 0.002) without significant group differences (Figure 4a). 

Aspartate aminotransferase (AST) levels as a function of time increased in all groups, with significant increases in the IPPV group (*p* = 0.043 BLH vs. T6) and the ULTVV group (*p* = 0.028); these did not significantly increase at T20 (*p* = 0.345 IPPV T6 vs. T20 and *p* = 0.753 ULTVV T6 vs. T20) (Figure 4b). Such an increase could not be observed in the sham group (*p* = 0.273 BLH vs. T6). Meanwhile, at T6 there was a significant difference between the respective intervention group and the sham group (*p* = 0.016 IPPV and *p* = 0.038 ULTVV); there were no significant differences found between the intervention groups (*p* = 1.0 T6 and *p* = 0.537 T20).

Post mortem analyses of liver tissues showed no significant differences in the total liver damage score (*p* = 0.078) (Figure 5a).

For the individual parameters of the liver damage scoring system, there were no significant differences between the intervention groups (Figure 5b). However, there was a significant difference between the sham and the respective intervention group regarding the parameter inflammation (*p* = 0.016 IPPV vs. sham, *p* = 0.010 ULTVV vs. sham). 

## 4. Discussion

### 4.1. Statement of Key Findings

To the best of our knowledge, our study was the first to investigate the effects of the ULTVV regimen during CPR on renal and hepatic end-organ damage in the short-term ROSC follow-up period. Our results substantiated that ULTVV, compared to standard IPPV, alleviates the adverse effects of cardiopulmonary resuscitation on creatinine clearance. Even so, ULTVV did show an increased renal histopathological score, which implies greater tissue damage. This contradiction can be explained by either a lack of validation of the EGTI score of the porcine kidney or the influence of the small group size. Furthermore, it is not inferior in terms of the restriction of the hepatic function and the hepatic histopathological damage in the short-term ROSC follow-up period. 

### 4.2. Explanation of the Selected Model

Since this investigation involves interventions at a time of extreme cardiocirculatory instability, this study protocol cannot be applied on human patients. In addition, the standardization of therapeutic processes is extremely difficult to implement in humans for ethical reasons. Even so, the chosen porcine ventricular fibrillation model is well established [22,23] and reliably reproducible [9]. In previous studies, this experimental setting could be used to investigate the effects of different ventilation strategies on hemodynamic parameters and ventilation-associated end-organ damage during CPR [24,25]. The juvenile pig is particularly suitable because its anatomy, physiology, and pathophysiological reactions are comparable to those of humans [26,27]; porcine lungs, in particular, are ideal for ventilation interventions because they are similar in size to the human lung [26]. Regarding drug therapy and recommendations on chest compressions, the currently valid recommendations on basic and advanced life support of the American Heart Association [2] and the European Resuscitation Council [1] were applied. Since the achievement of ROSC was essential for the research question of the present study, we added the application of vasopressin to increase the chances of ROSC [28]. 

In many cases, there are no reference values for laboratory parameters for piglets, especially renal function values and transaminases. Likewise, cut-off values for acute renal failure [29] as well as the EGTI scoring system and histopathological scoring of the liver were not validated in this application. For this reason, we decided to just report the values as they are. 

### 4.3. Connection with Previous Studies

Although the difference in ROSC rate was not significant, there was a bias in the ULTVV group in favor of a slightly better ROSC rate. This concurs with results of preliminary trials [6]. Here, too, was a non-significant bias in favor of the ULTVV group after four minutes of no-flow time compared to IPPV ventilation. Ruemmler et al. [6] surmised that this is caused by an improved ventilation-perfusion ratio and an improved venous return in the context of lower total intrathoracic pressures, presumably resulting in a better stroke volume [30]. 

Overall, our ROSC rate was well below the ROSC rate described by Wen et al. [31] after eight minutes of no-flow time after similar induction of ventricular fibrillation; that study reported a success rate of 100%. On the other hand, Rittenberger et al. [32] describe similar ROSC rates after a corresponding no-flow time with high-quality advanced life support, as in our study. The background concerning the discrepancies in these studies remains unclear.

Although the lactate levels showed no significant difference in the intervention groups, it is noteworthy that the IPPV group had the highest overall lactate increase after ROSC. An initially elevated lactate increase has been shown to correlate with an increased rate of kidney failure [7]. This fits with our finding of a reduced creatinine clearance observed in the IPPV group. 

The early increase in lactate that occurs in patients who experience cardiac arrest with subsequent resuscitation is an expression of general hypoperfusion. It is also described in the context of whole body ischemia reperfusion injury [33,34], that results in a relevant number of inflammatory mediators being released, which can lead to a sepsis-like syndrome [35] and distant end-organ damage [7,36]. This appears to be slightly more pronounced in the IPPV group. However, lactate clearance was sufficient and, after T6, was without significant differences to the sham group. This is consistent with the relatively low norepinephrine infusion rates that we had to administer to maintain target blood pressure, indicating relative clinical stability post-ROSC. 

The serum creatinine levels increased in all animals. A possible explanation would be that the chest compressions lead to a relevant creatinine release. However, this would not explain the increase in the sham group. Correspondingly, we also evaluated the creatinine clearance. The IPPV group showed a significantly poorer clearance than the sham group. According to Hasper et al. [37], increased rates of creatinine clearance are associated with a better neurological outcome post cardiac arrest. 

If one compares the clinical findings with the histopathological correlation, it is notable that all groups showed tubular damage in the EGTI score. This may represent the microscopic correlation to the elevated creatinine levels. In the CPR rat model, tubular damage correlated well with post-interventional loss of renal function or the rise of serum creatinine levels, respectively [38]. Even though tubular damage seems to be a typical finding in post-resuscitation pigs [39], our study cannot explain why this was also the case in the sham group. This might be attributed to a volume shift caused by the general anesthesia to which the animals of all three groups were exposed over a period of at least 20 hours due to the necessary sedation to avoid animal suffering. 

The fact that there were relevant differences in the liver scores between the sham and the respective intervention group is consistent with the observation that transaminases increased significantly in the intervention groups. This implies an injury of the cells from reduced blood flow, marked by hypoxia, lack of nutrients, and build-up of metabolic toxins during resuscitation or as a damage resulting from the global ischemia-reperfusion syndrome [8,40,41]. This study was not able to show an influence of the ventilation intervention on this histopathological scoring system. Aspartate aminotransferase (AST) is a relatively non-specific marker and can also be found in the myocardium [41]. Damage to the latter by resuscitation efforts can also lead to an increase in AST.

### 4.4. Clinical Significance 

This study provides evidence that ULTVV during CPR is beneficial because it mitigates renal functional restriction in the short-term post-ROSC course. We could not detect any adverse effects on liver function. In line with previous studies [6], which could show beneficial effects on cerebral integrity of this new ventilation mode, this supports the ongoing search for the optimal ventilation mode during CPR.

### 4.5. Limitations 

Our study shows the following limitations: (1) 20 h follow-up time post-ROSC only represents the early phase of post-resuscitation treatment. This circumstance is due to local legislation, according to which longer monitoring or post-interventional housing of the animals was not allowed. (2) In order to rule out confounding factors of a cyclical hormonal nature, we primarily worked with male pigs. Naturally, this may lead to our findings only being translationally applicable to female patients to a limited extent. (3) Our model works with an induced ventricular fibrillation. It is a well-established porcine CPR model [23]. However, it must be noted that, in the clinical setting, non-shockable rhythms are associated with a higher incidence of renal failure [7]. It is possible that the results of early kidney function impairments post-ROSC could have been more pronounced. (4) Due to low ROSC rates, only a few animals could be included for the relevant follow-up period, resulting in, among other things, high variations in statistical analysis. On the one hand, we attributed this to fatal injuries due to mechanical resuscitation devices. These serve well to ensure standardization of chest compressions but have limitations due to the unique anatomy of pigs with their wedge-shaped sternum [26,42]. Moreover, we attribute the relatively low ROSC rates to the eight minute no flow time. We chose this time to create more realistic environmental conditions, since immediate bystander CPR is not always applied in case of cardiac arrest [43,44]. However, to achieve reliable statistical analyses, more animals or a different resuscitation approach with a significantly higher ROSC rate would be necessary. (5) The scoring systems used have not been validated for the pig, nor for hepatic and renal end-organ damage as a result of resuscitation efforts. They can therefore only serve as an approximation.

## 5. Conclusions

ULTVV during CPR showed no difference in hemodynamic stability, such as norepinephrine requirement, in the short-term ROSC follow-up period when compared to standard ventilation IPPV. The creatinine clearance was found to be significantly more restricted for the standard therapy when compared to the sham group; this did not apply to ULTVV. Histopathologic findings of the kidney were unfavorable for ULTVV compared to the sham group; however, no clinical significance could be derived from this observation. Transaminases, lactate levels, and the histopathological evaluation of the liver with our newly developed liver scoring system showed no significant difference for ULTVV when compared to standard therapy. ULTVV could be an alternative to standard ventilation during resuscitation without adverse effects on the kidneys and liver. Further studies are required to address the unclear evidence of the international resuscitation guidelines on optimal ventilation during resuscitation and to review the clinical applicability and importance of alternative ventilation strategies during resuscitation.

## Figures and Tables

**Figure 1 biomedicines-11-00899-f001:**
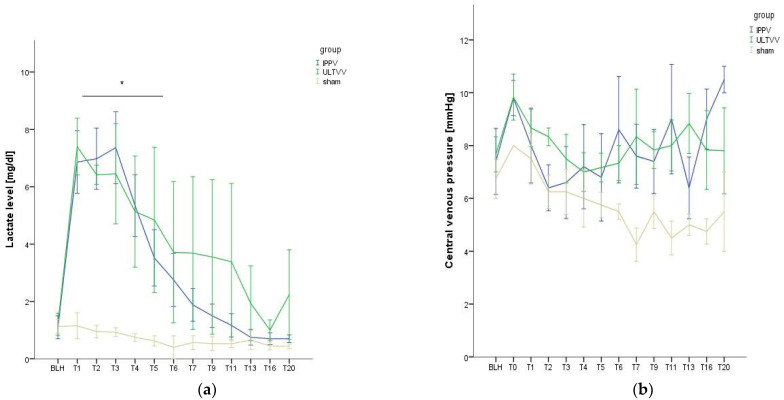
Hemodynamic stability. (**a**) Lactate levels as a function of time. (**b**) Central venous pressure as a function of time. Mean (±1 SE). * indicates *p* < 0.05 ULTVV and IPPV vs. sham treatment. BLH: baseline healthy. IPPV: Intermittent positive pressure ventilation. ULTVV: Ultra-low tidal volume ventilation. Tx: timepoint.

**Figure 2 biomedicines-11-00899-f002:**
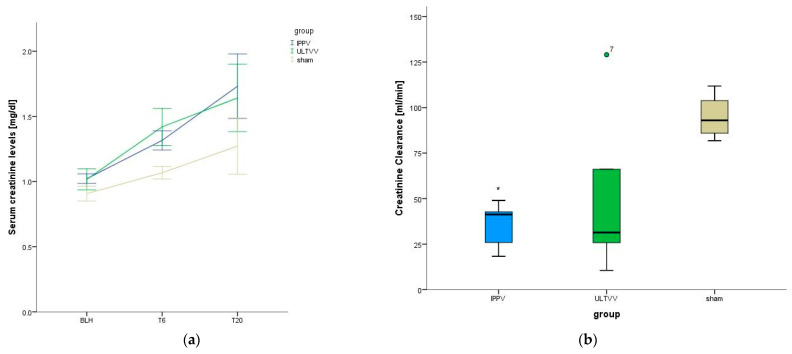
Kidney functional parameters. (**a**) Serum creatinine levels. (**b**) Creatinine clearance. Mean (±1 SE). * indicates *p* < 0.05 IPPV vs. sham treatment. BLH: baseline healthy. IPPV: Intermittent positive pressure ventilation. ULTVV: Ultra-low tidal volume ventilation. Tx: timepoint.

**Figure 3 biomedicines-11-00899-f003:**
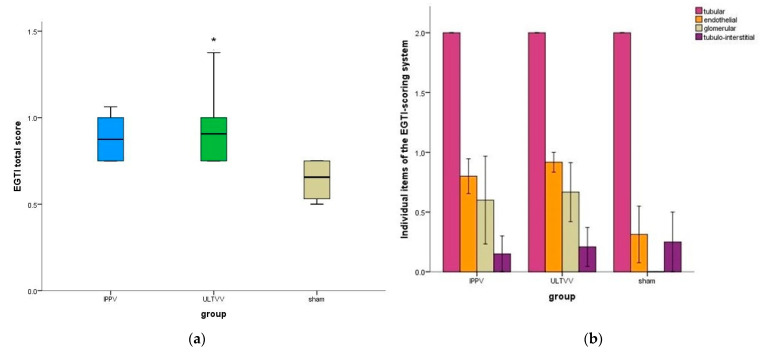
Histopathological scoring of the kidney (**a**) EGTI total score. (**b**) Individual items of EGTI score. Mean (±1SE). * indicates *p* < 0.05 ULTVV vs. sham treatment. IPPV: Intermittent positive pressure ventilation. ULTVV: Ultra-low tidal volume ventilation. EGTI score: endothelial, glomerular, tubular and tubulo-interstitial damage score.

**Figure 4 biomedicines-11-00899-f004:**
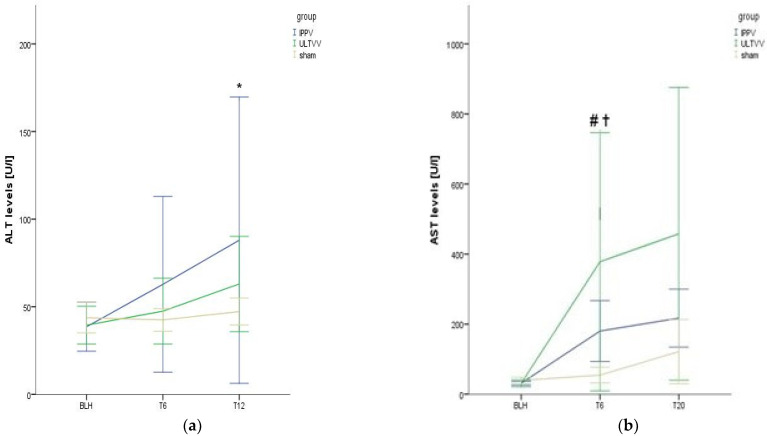
Levels of transaminases. (**a**) Alanine aminotransferase. (**b**) aspartate aminotransferase. Mean (±1 SE). * indicates *p* < 0.05 for elevated levels of alanine aminotransferase in all groups vs. baseline. # indicates *p* < 0.05 for IPPV vs. baseline and for IPPV vs. sham for T6. ✝ indicates *p* < 0.05 for ULTVV vs. baseline and for ULTVV vs. Sham for T6. IPPV: Intermittent positive pressure ventilation. ULTVV: Ultra-low tidal volume ventilation. U/L: Units per liter. ALT Alanine Aminotransferase. AST Aspartate Aminotransferase. Tx: timepoint.

**Figure 5 biomedicines-11-00899-f005:**
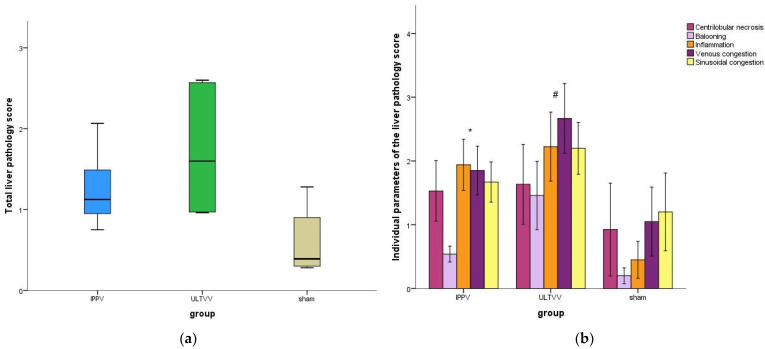
Histopathological scoring of the liver. (**a**) Total liver pathology score. (**b**) Individual parameters of the liver pathology score. Mean (±1 SE). * indicates *p* < 0.05 for IPPV vs. sham # indicates *p* < 0.05 for ULTVV vs. Sham. IPPV: Intermittent positive pressure ventilation. ULTVV: Ultra-low tidal volume ventilation.

**Table 1 biomedicines-11-00899-t001:** Ventilation mode during cardiopulmonary resuscitation via Intermittent Positive Pressure Ventilation (standard ventilation) and via Ultra-low Tidal Volume Ventilation (alternative ventilation). IPPV: Intermittent positive pressure ventilation. ULTVV: Ultra-low tidal volume ventilation. mL kg^−1^: Milliliters per kilogram. PEEP: Positive End Expiratory Pressure. FiO_2_: fraction of inspired oxygen. Min^−1^: per minute. n: numer of animals enlisted with sustained return of spontaneous circulation.

Ventilation Mode	Tidal Volume [mL kg^−1^]	PEEP [mbar]	FiO_2_	Respiratory Rate [Breaths min^−1^]	*n*
IPPV	8–10	5	1.0	10	5
ULTVV	2–3	5	1.0	50	6

**Table 2 biomedicines-11-00899-t002:** Hemodynamic parameters in group comparison. Mean (±SD). Group effects over time are analyzed by two-way-ANOVA and with post-hoc Bonferroni correction. There were no intergroup differences at baseline. n(IPPV) = 5; n(ULTVV) = 6; n(sham) = 4. * indicates *p* < 0.05 in intergroup comparison. HR: heart rate. MAP: mean arterial pressure. CVP: central venous pressure. NA: norepinephrine dosage. CI: cardiac index. Tx: timepoint. IPPV: intermittend positive pressure ventilation. ULTVV: ultra-low tidal volume ventilation.

Parameter	Group	Baseline	T6	T20
HR	IPPV	63 ± 12	85 ± 17	84 ± 50
	ULTVV	63 ± 15	93 ± 27	111 ± 59
	sham	62 ± 8	70 ± 18	65 ± 17
MAP	IPPV	67 ± 9	64 ± 6	77 ± 14
	ULTVV	73 ± 7	60 ± 6	57 ± 5
	sham	80 ± 5	80 ± 18 *	101 ± 8 *
CVP	IPPV	7 ± 3	9 ± 5	11 ± 1
	ULTVV	8 ± 2	7 ± 2	8 ± 4
	sham	7 ± 2	5 ± 1	5 ± 2
NA ^1^	IPPV	0 ± 0	0.069 ± 0.05	0.044 ± 0.07
	ULTVV	0 ± 0	0.187 ± 0.21	0.71 ± 1.17
	sham	0 ± 0	0 ± 0	0 ± 0
CI ^2^	IPPV	3.93 ± 1.87	2.81 ± 0.44	3.85 ± 1.64
	ULTVV	3.80 ± 1.13	3.63 ± 1.14	3.86 ± 1.08
	sham	3.50 ± 0.79	3.08 ± 0.82	3.97 ± 1.51

^1^ [µg kg^−1^ min^−1^], ^2^ [(L min^−1^)m^2−1^].

## Data Availability

All relevant data are presented in the manuscript. Specific data are available upon request.

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
