# Peer review of "The Influence of Ultra-Low Tidal Volume Ventilation during Cardiopulmonary Resuscitation on Renal and Hepatic End-Organ Damage in a Porcine Model"

_biomedicines, 2023, doi:10.3390/biomedicines11030899_

Round 1

Reviewer 1 Report

The authors reported the influence of ultra-Low tidal volume ventilation during  CRC on renal and hepatic end-organ damage in a porcine model. The submission requires major revision before any consideration taking into account the following points:-

1.      The number of analytical methods should be increased to have concrete evidence for

2.      The number of cases in each group should be highlighted.

3.      The authors reported statistical analysis, however, some figures missed standard errors.

4.      The variation in the recorded data of Figure 4 is high and should be explained.

5.      All abbreviations such as ‘ROSC’, ‘EGTI-score’….. should be fully defined when mentioned for the first time.

6.      The conclusion should be revised. For example, ‘ULTVV during CPR did not show any relevant disadvantages in the hemodynamic parameters in the short-term ROSC phase compared to standard ventilation IPPV.’ should be further supported with experimental data and similar studies.

7.      The language should be revised and typos should be corrected.

Reviewer 2 Report

I have read with interest the submitted paper, and I believe it can be of interest for the readers of Biomedicines.

I have only minor suggestions:

- Did authors assess the differences between repeated measures with validated "repeated measures" methods ? Please clarify in the text

- When defining the cohort in the results sections, I recommend to better specify the groups: author state that they performed "69 experiments", however, after excluding 11 animals (=58 animals), they obtained 3 groups of 4 (sham), ? (IPPV) and ? (ULTVV), and obtained ROSC in 5 IPPV and 6 ULTVV. Authors should define clearly the number in each group  both in the text and in Table 2.

- Were the differences observed between subgroups justified by subgroup numerosity ? In other words, did the sample size allow a sufficient statistical power ? I have some perplexities on the 4-cases sham subgroup.

- Table 2 is messed up, please improve the graphical presentation

Round 2

Reviewer 1 Report

The authors addressed most of the comments and the revised version can be accepted